# Developing a Clean Labelled Snack Bar Rich in Protein and Fibre with Dry-Fractionated Defatted Durum Wheat Cake

**DOI:** 10.3390/foods12132547

**Published:** 2023-06-29

**Authors:** Giacomo Squeo, Vittoria Latrofa, Francesca Vurro, Davide De Angelis, Francesco Caponio, Carmine Summo, Antonella Pasqualone

**Affiliations:** Department of Soil, Plant and Food Science (DISSPA), University of Bari Aldo Moro, Via Amendola, 165/a, I-70126 Bari, Italy; giacomo.squeo@uniba.it (G.S.); vittoria.latrofa@uniba.it (V.L.); francesca.vurro@uniba.it (F.V.); davide.deangelis@uniba.it (D.D.A.); francesco.caponio@uniba.it (F.C.); carmine.summo@uniba.it (C.S.)

**Keywords:** cereal based, texture, upcycling, ready to eat food, design of experiments, texture analysis, nutritional composition

## Abstract

The shift towards a vegetarian, vegan, or flexitarian diet has increased the demand for vegetable protein and plant-based foods. The defatted cake generated during the extraction of lipids from durum wheat (*Triticum turgidum* L. var. *durum*) milling by-products is a protein and fibre-containing waste, which could be upcycled as a food ingredient. This study aimed to exploit the dry-fractionated fine fraction of defatted durum wheat cake (DFFF) to formulate a vegan, clean labelled, cereal-based snack bar. The design of experiments (DoEs) for mixtures was applied to formulate a final product with optimal textural and sensorial properties, which contained 10% DFFF, 30% glucose syrup, and a 60% mix of puffed/rolled cereals. The DFFF-enriched snack bar was harder compared to the control without DFFF (cutting stress = 1.2 and 0.52 N/mm^2^, and fracture stress = 12.9 and 9.8 N/mm^2^ in the DFFF-enriched and control snack bar, respectively), due to a densifying effect of DFFF, and showed a more intense yellow hue due to the yellow–brownish colour of DFFF. Another difference was in the caramel flavour, which was more intense in the DFFF-enriched snack bar. The nutritional claims “low fat” and “source of fibre” were applicable to the DFFF-enriched snack bar according to EC Reg. 1924/06.

## 1. Introduction

One third of food produced for human consumption is lost or wasted worldwide, accounting for about 1.3 billion tons annually [1]. Alongside this, food supply chains generate a large amount of valuable by-products that still could be exploited and valorised for human nutrition. As for the supply chain of durum wheat (*Triticum turgidum* L. var. *durum*), during the milling process, a by-product is generated composed of germ, bran, and debranning fractions [2]. Being rich in high biological value protein, minerals, fibre, and lipids, this by-product can be further exploited [3,4]. Indeed, the extraction of the lipid fraction, rich in vitamins and essential fatty acids, has been proposed [5], and durum wheat oil has been used for the preparation of biscuits and focaccia [6,7]. The lipid extraction, however, generates a second by-product, namely, the defatted durum wheat cake. To upcycle the defatted cake, after stripping and micronisation, the dry-fractionation process can be applied, allowing a coarse fraction rich in starch and fibre and a fine fraction rich in protein to be obtained [8]. The coarse fraction of dry-fractionated cereal flours has already been proposed as an ingredient in different food products, such as meat analogues and spaghetti [9,10]. In addition, the fine fraction can also be used to fortify foods [11] with the advantage of being derived by a more sustainable method of protein enrichment compared to a wet concentration, which uses an alkaline extraction followed by isoelectric precipitation [8].

The sustainability of food systems is a hot topic increasingly considered by researchers, national/international organizations, and consumers. Additionally, the recovery of protein from vegetable waste and by-products matches the general request to replace animal protein with alternative plant protein [12]. In fact, the shift towards a vegetarian, vegan, or flexitarian diet has increased the demand for vegetable protein and plant-based foods [13,14]. The reduction of animal protein is associated with well-being and health, both being drivers that play a pivotal role in influencing the food choices of many consumers who integrate nutrition with sport activities [15]. To support this healthy lifestyle, consumers increasingly choose foods rich in protein and fibre, usually marketed in the form of snack bars [16,17]. In addition, the fast pace of life further concurs to the “snacking” effect, mostly based on the consumption of snack bars.

Given that sweet snacks are considered unhealthy due to the presence of high levels of sugars, food companies are releasing snacks formulated in a healthier way, incorporating protein isolates to increase protein content [18] and preferring clean labelled formulations [19,20]. Though specific regulations/legislations of clean labels do not exist, and a clear definition has not been established [21], this new trend in food products, mostly referring to the absence of additives, has been taken up by a multitude of food industry stakeholders [22]. Indeed, many additives can be included in the formulation of snack bars. Among the most used, sorbitol and polyols, responsible for a laxative effect, are sugar replacers, while lecithin is the most common emulsifier, and glycerol is added as a gelatinizing agent, followed by colouring agents and artificial flavours to improve the sensory properties [23]. Moreover, saturated fats, involved in cholesterol increase and cardiovascular disease, are reported among the variety of ingredients commonly used in snack bars [23].

Over the past few years, several studies have been conducted concerning the re-use of food industry by-products (such as the bran of black rice and corn, banana peel powder, sunflower meal, and jackfruit seed flour) in the formulation of snack bars [20,24,25,26,27]. These studies have confirmed the increasing interest towards this food category, highlighting its suitability as a recipient for fortification and functionalization. However, the re-use of defatted durum wheat cake, which can be easily dry-fractionated to obtain a protein-rich flour, has not been considered so far.

In this framework, this study aimed at exploiting the dry-fractionated fine fraction of defatted durum wheat cake (DFFF) to formulate a vegan, clean labelled, cereal-based snack bar, evaluating the influence of the ingredients on the textural, sensory, and nutritional properties of the final product.

## 2. Materials and Methods

### 2.1. Basic Ingredients

DFFF was provided by Casillo Next Gen Food srl (Corato, Italy). Its preparation was as follows: The durum wheat milling by-product was composed of a mix of germ, bran, and debranning fractions, and it was submitted to the extraction of the oily fraction using *n*-hexane, as described in the study by Squeo et al. (2022) [5]. The remaining cake was micronized using an impact mill (UPZ 100, Hosokawa-Alpine, Augusta, Germany). The mill speed was set at 15,000 rpm and the feed rate at 3 kg h^−1^. Then, an air classifier (ATP 50, Hosokawa-Alpine, Augusta, Germany) was used to dry-fractionate the micronized cake flour into coarse and fine fractions. The speed of the grader wheel was set at 3000 rpm and the air flow rate at 50 m^3^ h^−1^. The fine fraction was recovered, having an *n*-hexane residual content <5 ppm, i.e., food grade. The chemical composition of DFFF is shown in Figure 1.

Puffed hulled millet (Vivibio, Villareggia, Italy), 5-grain rolled cereals (20% oat, 20% barley, 20% rice, 20% rye, 20% wheat) (Fiorentini, Troffarello, Italy), and puffed spelt (Selex, Trezzano, Italy) were purchased from a local retailer.

### 2.2. Experimental Design and Preparation of the Snack Bars

The design of experiments (DoEs) for mixtures [28] was applied to formulate a product with optimal textural and sensorial properties. Three factors were considered, which were the three main ingredients, namely, DFFF, glucose syrup, and a cereal mix composed of puffed hulled millet, 5-grain rolled cereals, and puffed spelt (35:40:25 *w*/*w*/*w*). The quantitative ranges were 10–50%, 30–60%, and 20–60% for DFFF, glucose syrup, and cereal mix, respectively. Then, from all the possible mixtures in the defined experimental domain, 8 unique formulations were chosen (Table 1) by a D-optimality criterion [29] to model a special cubic response.

To prepare the snack bars, glucose syrup (Ambrosio, Striano, Italy) and DFFF were manually mixed and heated in a pan on an induction heater (G3 Ferrari, Rimini, Italy) at the power of 2000 W until the complete formation of a cream was reached (about 5 min). Then, the cereal mix was added and manually mixed to distribute it homogeneously into the cream, followed by pouring into a silicone mould (FantasyDay, Changan Town, China) having six 2 cm thick cavities of 11.5 × 3.8 cm, and cooking in a ventilated oven (Smeg, Guastella, Italia) for 5 min at 120 °C. The snack bars were allowed to cool down at room temperature (20 ± 1 °C) and then extracted from the moulds.

Three snack bars for each formulation of the experimental design were prepared (*n* = 3). A control snack bar, without DFFF, was prepared with 33.3 g 100 g^−1^ of glucose syrup and 66.6 g 100 g^−1^ of cereal mix.

### 2.3. Determination of the Chemical Composition

Protein content (N × 5.7) was determined according to the method AOAC 979.09 [30]. The lipid content was determined using a Soxhlet apparatus, according to the method AOAC 945.38F [30], using diethyl ether as the extraction solvent. The total dietary fibre was analysed by the enzymatic–gravimetric procedure as described in the method AOAC 985.29 [30]. Moisture and ash content were determined according to the AOAC methods 925.10 and 923.03 [30], respectively. The available carbohydrate content was determined by difference, subtracting the content of protein, lipid, fibre, moisture, and ash to 100. The energy value was calculated by using Atwater general conversion factors. All determinations were carried out in triplicate.

### 2.4. Texture Analysis

The textural properties of the snack bars were determined by a cutting test and a three-point bending test. A texture analyser, equipped with a 1 kN load cell (Z1.0 TN, Zwick Roell, Ulm, Germany), was used for both tests.

The cutting test was carried out as described in Costantini et al. [31] with some modification. The sample (11.5 × 3.8 × 2 cm, L × W × H) was positioned on the base in order to be cut by the 50 mm length blade on its short side positioned 25 mm from the surface of the sample. The blade descended at a speed of 1 mm/s until the sample was cut. The cutting stress, i.e., the maximum force applied on the surface until the sample was completely cut (N/mm^2^), and the deformation until the cut (mm) were measured.

Three-point bending was carried as reported in Pasqualone et al. [32] with a few modifications. The texture analyser was equipped with a 1 kN load cell, a metal breaking probe, and two metal supports, 6 cm apart from each other, to hold the whole sample (11.5 × 3.8 × 2 cm, L × W × H). The sample was placed on the two supports, and the probe (positioned 25 mm above the sample) descended on it at a speed of 5 mm/s. The fracture stress (N/mm^2^), i.e., the maximum force required to break the sample, and the deformation until rupture (mm) were measured.

Data acquisition was accomplished using TestXPertII v3.41 software (Zwick Roell, Ulm, Germany). The analysis was carried out in triplicate.

### 2.5. Sensory Analysis

The sensory features were evaluated by a panel consisting of 8 trained members aged between 26 and 56 years. The panellists did not suffer from any food intolerances or allergies, received information on the objectives of the study, and provided written informed consent.

At first, the snack bars corresponding to the 8 different formulations (Table 1) were evaluated on a 0–9 score range (0 = minimum; 9 = maximum) for their stickiness (a critical parameter for these products) and acceptability. The results of this evaluation, together with the textural data, allowed the optimal snack bar to be selected. Then, both the optimal and the control snack bar were submitted to Quantitative Descriptive Analysis (QDA) for a detailed sensory profiling. Eight sensory descriptors were evaluated on a 0–9 score range (0 = minimum; 9 = maximum intensity) regarding visual appearance (homogeneity of the distribution of cereals, brightness, yellowish colour), flavour (cereal-like, toasted, caramel and abnormal), and taste (sweet).

### 2.6. Colour Analysis

The snack bars had a highly heterogeneous colour that could not be analysed directly with a colourimeter, and, for this reason, image analysis and colour elaboration was carried out. An RGB image was collected using a Canon EOS 600D camera equipped a Canon EF-S 18–200mm f/3.5–5.6 IS lens (Canon Inc., Tokyo, Japan) at a 50 cm distance between the lens and the sample. The optimal and control snack bars, placed inside a white plastic box, sized 49 × 51 × 50 cm, with a black base, were photographed from above. The image (2193 × 2683 pixels) was imported into MATLAB R2021a (The MathWorks Inc., Natick, MA, USA) and processed as follows: After masking and background removal, carried out using the HYPER-Tools toolbox (freely available at www.hypertools.org, accessed on 20 April 2023) [33], the image was converted from RGB colour space to *L***a***b** colour space, thus obtaining the respective values of brightness and red and yellow indices per pixel. Then, the absolute frequency distribution and the cumulative frequency distribution of the pixels were calculated per each *L***a***b** index to highlight differences in the colour characteristics of the snack bars.

### 2.7. Statistical Analysis

The mixture experimental design was set up and analysed using CAT (Chemometric Agile Tool) software [34], freely downloadable from http://gruppochemiometria.it/index.php/software, accessed on 20 April 2023. All data were expressed as the mean ± standard deviation (*n* = 3). One-way analysis of variance (ANOVA) was performed, followed by the Tukey test for multiple comparisons using Minitab 17 (Minitab Inc., State College, PA, USA).

## 3. Results and Discussion

### 3.1. Identification of the Optimal Formulation

Figure 2 depicts the contour plots of the cutting stress (A), the fracture stress (B), the sensory acceptability (C), and the stickiness (D) of the experimental snack bars in the defined domain. DoEs allowed us to reach a comprehensive understanding of the ingredients’ effects on the snack bar properties and to select the better formulation. A snack bar, indeed, is a mixture of ingredients that substantially retain their original physical characteristics, which are generally very different from each other, so their relative quantities greatly influence the features of the final product.

The cutting and fracture stress, determined by measuring the maximum force required to cut or break the sample, respectively (Figure 2A,B), were considered because they are known to be related to sensory firmness and stickiness to the teeth [35], which are important characteristics of snack bars. The fitting of the responses, however, was not excellent, showing in both the cases an R^2^ value equal to 0.6. This was likely due to the high physical inhomogeneity of this kind of product, which brought great variability in the responses, making the modelling more difficult. Nonetheless, from the contour plots, useful information about the effects of the ingredients on the mechanical features of the snack bar could be caught. Indeed, the cutting and fracture stress values, and thus the hardness of the snack bars, progressively increased by raising the proportion of X2 (glucose syrup). Therefore, the highest amount of glucose syrup gave rise to very hard bars, as observed close to the X2 vertex. On the other hand, the experimental region close to the X1 vertex corresponded to the worst results due to a total lack of consistency. The snack bars in this region (trials 1 and 8) were impossible to shape properly, showing a powdery and poor structure (Figure 3). Experiments 1 (50% DFFF, 30% glucose syrup, 20% cereals) and 8 (30% DFFF, 30% glucose syrup, 40% cereals) highlighted that the snack bar could not be formed when DFFF was ≥ glucose syrup. Indeed, DFFF was rich in protein and fibre, so its increase would be nutritionally valuable [36], but being a powdery ingredient, an excessive amount disrupted the structure of the snack bar, especially when the amount of glucose syrup was too low. At the same time, high amounts of glucose syrup excessively increased the hardness. By increasing the amount of the cereal mix (X3), a moderate and positive effect on the structure was observed. Therefore, for the textural features, the best experiments were those in the centre of the domain (point 7), at the X3 vertex (point 4), and along the edge of X3–X2 (points 3 and 5) (Figure 2A,B).

Together with the textural properties, two other parameters were chosen for the selection of the optimal formulation, i.e., sensory acceptability, which is fundamental for food products in general, and stickiness, which is important especially for the snack bars. The responses for these two parameters are depicted in Figure 2C,D, respectively. In this case, the fitting of the responses was better than the textural parameters, showing an R^2^ value equal to 0.9 for the sensory acceptability and 0.7 for the stickiness, both markedly influenced by variations in the ingredient ratio. The stickiness was very high in the upper part of the X3–X2 edge (experiments 3 and 5). The highest sensory acceptability and the lowest stickiness were observed in the experimental region close to the X3 vertex (point 4).

In light of these results, the most suitable trade-off was considered to be the formulation corresponding to the X3 vertex (experiment 4), i.e., 10% DFFF, 30% glucose syrup, and 60% cereals. This well-structured bar, neither too sticky nor too hard, was the optimal one and was then further characterized for its nutritional, textural, and sensorial features.

### 3.2. Nutritional, Textural, and Sensorial Characterization

Table 2 shows the results of the nutritional composition of the DFFF-enriched snack bar, compared with the control snack bar, prepared without DFFF.

The addition of DFFF caused a significant reduction of the moisture and an increase in protein content with respect to the control. These results were expected, DFFF being an ingredient high in protein and low in moisture. In particular, the increase in protein highlighted the effectiveness of using a dry-fractionated ingredient considering that studies involving the addition of other cereal by-products, such as black rice bran and maize bran [20,26,39], did not achieve a significant increase in protein content. One DFFF-enriched snack bar (weight = 33 g) covered 6.4% of the daily reference intake of protein, while the control snack bar provided only 4.3% of the reference intake. The consumption of plant-based protein has a positive effect on health, since it minimizes cardiometabolic risk factors and is inversely correlated with hypertension, obesity, and insulin resistance [40].

DFFF is also rich in fibre (Figure 1). Therefore, another positive result of the incorporation of DFFF was the enrichment in total dietary fibre (Table 2), recommended to improve the metabolism of gut microbiota and to reduce the risks connected with diabetes, obesity, and dyslipidaemia [41]. Moreover, the reached amount of fibre makes possible the labelling of the DFFF-enriched snack bar as a “source of fibre” according to the current rules [42], as it meets the minimum requirement of 3 g of fibre per 100 g of product. This amount was higher than the level reached by adding other by-products of the food industry, such as pumpkin seed flour or carrot powder [43,44]. Specifically, a single DFFF-enriched snack bar (33 g) provided 4.2% of the daily intake recommended by the EFSA for dietary fibre (i.e., 25 g) [38], while the control bar provided 3.0% of the reference intake.

The lipid content did not show a significant difference between the two snack bars. The observed amount makes possible the labelling of them as “low fat” according to the EC Reg. 1924/2006 [42] (no more than 3 g of fat in 100 g). The observed lipid content was considerably lower than in snack bars formulated with other by-products, such as tilapia frames or roasted rice bran [39,45].

Carbohydrates remained the main constituents of the snack bars, but they were significantly lower in the DFFF-enriched bar than in the control.

The two types of snack bars showed very similar energy values, providing 6% of the daily reference value of 2000 kcal [37].

As for the texture analysis of the snack bars, both cutting and bending tests, which helped also in selecting the optimal formulation, highlighted significant textural differences between the DFFF-enriched snack bar and the control, with the former being harder than the latter. The results are shown in Table 3.

The observed increase in cutting and fracture stress was likely due to the densifying effect of DFFF when added to glucose syrup, leading to a thick cream that made the snack bar become very compact and hard. These results were correlated with the higher extent of deformation observed in the DFFF-enriched bar with respect to the control. This indicated a greater effort needed to chew the DFFF-enriched snack bar compared to the control and, more in general, evidenced the ability of the new formulation to tolerate higher stress. Foods that need longer chewing could reduce the appetite and then also the energy intake [46]. Therefore, a harder texture is desirable from a nutritional point of view. Zulaikha et al. [45] and Silva et al. [47] observed a similar increase in hardness following the addition of powdery ingredients, such as tilapia frames and marolo pulp flour, to snack bars.

As for the colour of the snack bars, high heterogeneity was found along their surfaces, as expected in this kind of product (Figure 3). As a consequence, a direct colorimeter evaluation was not sufficiently reliable, justifying the need for imaging approaches to better evaluate the product’s colour. Figure 4A shows the image of the snack bars reconstructed according to the *L***a***b** values of each pixel. In both the snack bars, the surface of the cereal grains showed higher values of brightness (*L**), which could be explained by the shiny coating formed by the glucose syrup. The pixel distribution of the *L** value (Figure 4B) was bimodal, i.e., presented two main peaks, indicating a strong contrast within the surface brightness in different parts of the snack bar due to the heterogeneity of the assembled ingredients. The curves of the cumulative distribution showed that the control bar had a higher percentage of pixels with high *L** values with respect to the DFFF-enriched snack bar, supporting the empirical observation (Figure 4A) that the former was, overall, brighter than the latter.

High rates of negative *a** values were recorded in both snack bars (Figure 4A), indicating a general trend toward the greenish colour, with some surface portions characterized by positive values and thus reddish features. The pixel distribution (Figure 4B) was very similar between the two snack bars, showing minor differences in the whole red–green feature, but with a higher proportion of reddish pixels in the DFFF-enriched snack bar with respect to the control.

A remarkable difference was observed for the *b** index, with the DFFF-enriched snack bar clearly presenting a more intense yellow hue (i.e., positive values) with respect to the control bar (Figure 4A). The DFFF ingredient, indeed, was characterized by a yellow–brownish colour that was reflected in a more yellow bar. This empirical observation was supported by the frequency distribution (Figure 4B) that clearly showed a higher proportion of pixels having high *b** values in the DFFF-enriched snack bar with respect to the control.

Overall, it could be concluded that the DFFF-enriched snack bar was slightly less bright but redder and yellower than the control bar.

The QDA results are reported in Table 4. The visual evaluation revealed statistically significant differences in the colour, with the DFFF-enriched snack bar perceived as more yellow than the control, in agreement with the instrumental analysis. Brightness was also in line with the instrumental results because the DFFF-enriched bar was significantly less bright than the control, probably because the addition of DFFF diminished the glossy appearance of the surface.

From a practical point of view, the corroboration of the results of the instrumental colour evaluation and of the sensory evaluation suggests that image processing technology may be implemented on the production process for quality control procedures, considering also that colour determination with classical colorimeters is challenging for such heterogeneous products, and the sensory evaluation requires trained panellists and is time consuming. On the contrary, no significant differences emerged in the homogeneity of the distribution of cereals.

Flavour attributes were ascribed to the typical characteristic of cereals and/or connected to the cooking step. The addition of the DFFF did not affect the cereal-like note. The flavour of caramel, instead, was significantly more intense in the DFFF-enriched snack bar than in the control, probably due to the thermal effect exerted by the hot glucose syrup on DFFF during the preparation of the snack bar. Heat treatments of bran and similar materials are known to develop volatile compounds with sweet and caramel-like aromas [48]. Indeed, also the toasted note was perceived more intensely in the DFFF-enriched snack bar, but the difference with the control was not statistically significant. No abnormal smells were detected in either samples. The taste evaluation showed that both the snack bars had an equally perceived medium sweet taste, imputable to the glucose syrup.

## 4. Conclusions

The use of DFFF as a food ingredient could favour the circularity of the durum wheat supply chain while responding to the demand for healthy and ready-to-eat food products expressed by modern consumers. The outcomes of this study demonstrated that the development of an innovative snack bar enriched with DFFF is a feasible strategy to this end.

DFFF should not be added in quantities higher than 10%, and in any case must be lower than the amount of glucose syrup in order to ensure an optimal structure and adequate sensory acceptability of the snack bar. Nonetheless, thanks to the fortification with DFFF, the proposed snack bar showed improved nutritional characteristics, with enhanced protein content. Noteworthily, the optimized formulation can be labelled as “low fat” and a “source of fibre” according to the EC Reg. 1924/06. Furthermore, being exclusively made of plant-based ingredients, without additives, the DFFF-enriched snack bar was a clean-labelled food product and met the expectations of vegetarian and vegan consumers.

This study could pave the way for the formulation of other DFFF-enriched snack bars, for example, by incorporating nuts and/or dehydrated fruit, to offer the consumer a wide choice of flavours and tastes while meeting the current trends in sustainable food consumption.

## Figures and Tables

**Figure 1 foods-12-02547-f001:**
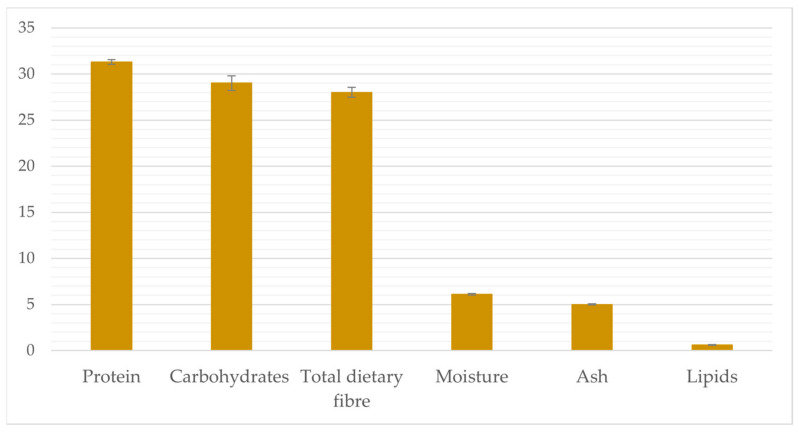
Proximate composition (g 100 g^−1^) of dry-fractionated fine fraction of defatted durum wheat cake (DFFF).

**Figure 2 foods-12-02547-f002:**
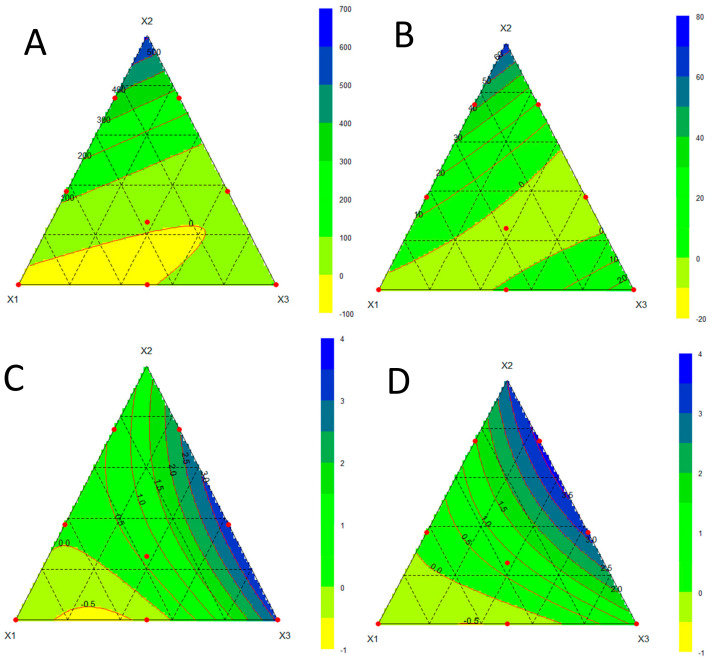
Contour plots of the cutting stress (N/mm^2^, (**A**)), fracture stress (N/mm^2^, (**B**)), sensory acceptability (**C**), and stickiness (**D**) of the experimental snack bars. X1 = dry-fractionated fine fraction of defatted durum wheat cake (DFFF); X2 = glucose syrup; X3 = puffed/rolled cereal mix. Red point represents the experiments carried out.

**Figure 3 foods-12-02547-f003:**
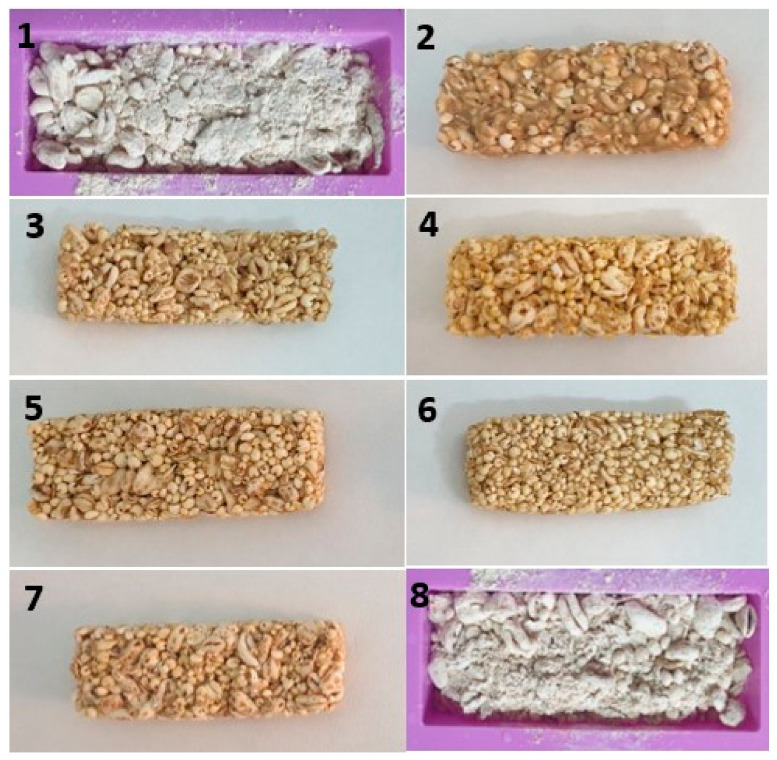
Experimental snack bars formulated with dry-fractionated fine fraction of defatted durum wheat cake (DFFF), glucose syrup, and puffed/rolled cereal mix. Numbers 1–8 correspond to the formulations reported in Table 1.

**Figure 4 foods-12-02547-f004:**
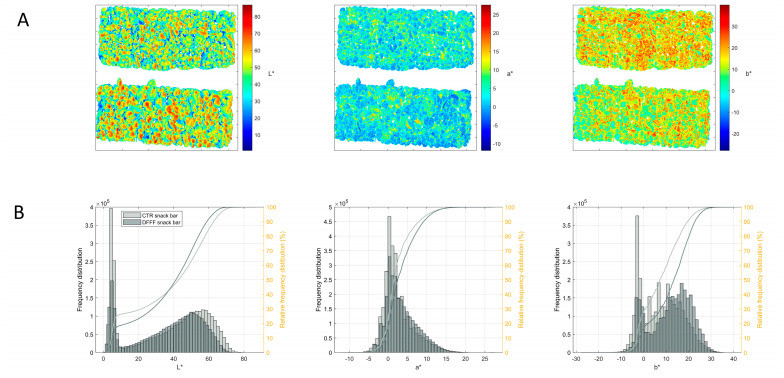
(**A**) *L**, *a**, and *b** reconstructed images of the DFFF-enriched snack bar (snack bar on the top of each image) and the control snack bar (snack bar on the bottom of each image). (**B**) Pixel absolute frequency distribution (bars) and cumulative frequency distribution (lines) of the control snack bar (light grey) and of the DFFF-enriched snack bar (dark grey) for *L**, *a**, and *b**.

**Table 1 foods-12-02547-t001:** List of the experiments performed.

Experiment	Dry-Fractionated Fine Fraction of Defatted Durum Wheat Cake—X1 (g 100 g^−1^)	Glucose Syrup—X2 (g 100 g^−1^)	Puffed/Rolled Cereal Mix—X3 (g 100 g^−1^)
1	50	30	20
2	20	60	20
3	10	60	30
4	10	30	60
5	10	45	45
6	35	45	20
7	25	40	35
8	30	30	40

**Table 2 foods-12-02547-t002:** Nutritional composition of the experimental snack bars. DFFF = dry-fractionated fine fraction of defatted durum wheat cake.

Parameter	DFFF-Enriched Snack Bar			Control Snack Bar		
	Amount per 100 g	Amount per Unit (33 g)	% Daily Value per Unit (33 g) *	Amount per 100 g	Amount per Unit (33 g)	% Daily Value per Unit (33 g) *
Moisture (g)	8.3 ± 0.11 ^b^	2.7	-	9.3 ± 0.17 ^a^	3.2	-
Protein (g)	9.7 ± 0.24 ^a^	3.2	6.4	6.5 ± 0.08 ^b^	2.3	4.3
Total dietary fibre (g)	3.2 ± 0.20 ^a^	1.1	4.2	2.3 ± 0.31 ^b^	0.8	3.0
Lipids (g)	1.4 ± 0.09 ^a^	0.5	0.7	1.4 ± 0.04 ^a^	0.5	0.7
Carbohydrates (g)	76.4 ± 0.36 ^b^	25.2	9.8	80.1 ± 0.63 ^a^	26.4	10.2
Energy value (kcal)	363	120	6.0	364	120	6.0

* Energy, protein, lipids, and carbohydrates were calculated according to the daily reference intakes reported in the Annex XIII of the EU Reg. 1169/2011 [37]; total dietary fibre was calculated according to the daily reference intake reported in the EFSA recommendation [38]. Different letters indicate significant differences at *p* < 0.05.

**Table 3 foods-12-02547-t003:** Texture analysis of the experimental snack bars. DFFF = dry-fractionated fine fraction of defatted durum wheat cake.

Parameter	Type of Snack Bar	
	DFFF-Enriched	Control
Cutting test		
Cutting stress (N/mm^2^)	1.2 ± 0.1 ^a^	0.52 ± 0.2 ^b^
Deformation (mm)	16.3 ± 0.29 ^a^	14.7 ± 0.56 ^b^
Three point-bending test		
Fracture stress (N/mm^2^)	12.9 ± 3.9 ^a^	9.8 ± 4.09 ^b^
Deformation (mm)	3.6 ± 0.8 ^a^	2.5 ± 1.07 ^b^

Different letters indicate significant differences at *p* < 0.05.

**Table 4 foods-12-02547-t004:** Results of the quantitative descriptive sensory analysis of the experimental snack bars. DFFF = dry-fractionated fine fraction of defatted durum wheat cake.

Sensory Parameter	Type of Snack Bar	
	DFFF-Enriched	Control
Visual		
Yellowish colour	6.1 ± 1.2 ^a^	4.0 ± 0.9 ^b^
Brightness	5.0 ± 1.1 ^b^	7.1 ± 0.8 ^a^
Homogeneity of the distribution of cereals	6.8 ± 0.4 ^a^	7.0 ± 0.6 ^a^
Flavour		
Cereal-like	6.8 ± 0.9 ^a^	6.8 ± 1.3 ^a^
Toasted	4.7 ± 1.2 ^a^	3.8 ± 1.0 ^a^
Caramel	4.1 ± 0.7 ^a^	2.2 ± 1.1 ^b^
Abnormal odours	0.2 ± 0.4 ^a^	0.3 ± 0.8 ^a^
Taste		
Sweetness	4.5 ± 0.5 ^a^	4.1 ± 0.5 ^a^

Different letters indicate significant differences at *p* < 0.05.

## Data Availability

The data used to support the findings of this study can be made available by the corresponding author upon request.

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
