# Peer review of "Developing a Clean Labelled Snack Bar Rich in Protein and Fibre with Dry-Fractionated Defatted Durum Wheat Cake"

_foods, 2023, doi:10.3390/foods12132547_

Round 1
Reviewer 1 Report
Since a clean label is an emphasized attribute of the snack bar elaborated in the manuscript, it would be interesting to dedicate a paragraph in the Introduction section to the topic of clean label products with more details about which additives are used in snack bar production and why is it important to avoid them.
Paragraph 2.3 Pages 116-126: Did the formula for calculating available carbohydrates include ash content when subtracting from 100?
Paragraph 2.4, pages 128-144. Explain in more detail how was cutting and fracture stresses derived from the Force-time curves? Was it determined as an area under the curve or maximum force at deformation? Since the result is expressed as stress, how was the sample prepared for the analysis and how was sample geometry measured for stress calculation?
In the Abstract, it is stated that the nutritional claims "rich in protein i.e. fibre" are eligible for the snack bar. But, in R&D lines 255-277, only "source of fibre" is mentioned as well as "low fat". This should be checked and harmonized.
The Conclusion should explicitly state what is the nutritional improvement of the optimised bar.
Author Response
We thank the reviewers for their useful comments. All the suggested observations have been addressed and the revisions have been highlighted in RED in the manuscript. Following the point-to-point response to Reviewer.
Reviewer #1:
- Since a clean label is an emphasized attribute of the snack bar elaborated in the manuscript, it would be interesting to dedicate a paragraph in the Introduction section to the topic of clean label products with more details about which additives are used in snack bar production and why is it important to avoid them.
Answer: Thank you for the interesting observation. According to the suggestion, the introduction has been expanded by adding a paragraph on the clean label and by considering a recent review regarding the ingredients used for the preparation of snack bars, citing also the possible undesirable effects for human health. Consequently, the number of references has been updated (Lines 60-68).
- Paragraph 2.3 Pages 116-126: Did the formula for calculating available carbohydrates include ash content when subtracting from 100?
Answer: Yes, it included ash content, thanks for noting the typo. The section 2.3 has been all checked and other small mistakes (such as a sentence to be deleted on water activity determination, not pertaining to this manuscript, due to a residual copy-paste) have been amended (lines 130-133).
- Paragraph 2.4, pages 128-144. Explain in more detail how was cutting and fracture stresses derived from the Force-time curves? Was it determined as an area under the curve or maximum force at deformation? Since the result is expressed as stress, how was the sample prepared for the analysis and how was sample geometry measured for stress calculation?
Answer: In paragraph 2.4 we have better explained how the samples was utilized during the analysis, by reporting the dimensions of the snack bars and of the blade (Lines 141-142; 149, 151). The cutting and fracture stresses correspond to the maximum force required to cut and break the sample of the given dimensions, also considering the blade dimensions for cut and the distance between the support for the three-point bending test. Those values are reported in the software before the analysis to allow the calculation of the parameters.
- In the Abstract, it is stated that the nutritional claims "rich in protein i.e. fibre" are eligible for the snack bar. But, in R&D lines 255-277, only "source of fibre" is mentioned as well as "low fat". This should be checked and harmonized.
Answer: According to the suggestion, the abstract has been harmonized with the R&D (line 23). Thanks for noting.
- The Conclusion should explicitly state what is the nutritional improvement of the optimised bar.
Answer: According to the suggestion, the Conclusions paragraph has been implemented with the information about the nutritional improvement (lines 385-387).

Reviewer 2 Report
It is a paper that is very careful; however, some changes are necessary so that the paper reflects the good experimental work. Here are my comments about it:
Line 52: Regarding the reduction of animal protein and its replacement by vegetable proteins, it would be interesting to point out some of the works that include legumes as a source of protein in snacks such as
Arribas, C., Cabellos, B., Cuadrado, C., Guillamón, E., & Pedrosa, M. M. (2019). Extrusion effect on proximate composition, starch and dietary fibre of ready-to-eat products based on rice fortified with carob fruit and bean. LWT, 111, 387–393. https://doi.org/10.1016/j.lwt.2019.05.064
Ciudad-Mulero, M., Fernández-Ruiz, V., Matallana-González, M. C., & Morales, P. (2019). Dietary fiber sources and human benefits: The case study of cereal and pseudocereals. Advances in Food and Nutrition Research, 90, 83–134. https://doi.org/10.1016/bs.afnr.2019.02.002
Figure 1: please, include in the graph the bars of the standard deviation of determinations.
Table 2: Perhaps it would be interesting to also include the energy intake of a bar and include the % that covers daily needs (at least in fiber). This information can be included in the text as well.
Perhaps more emphasis could be placed on the correlation between the values ​​evaluated technologically and the characterization made by the experts
Minor editing of English language required
Author Response
We thank the reviewers for their useful comments. All the suggested observations have been addressed and the revisions have been highlighted in RED in the manuscript. Following the point-to-point response to Reviewer.
Reviewer #2:
It is a paper that is very careful; however, some changes are necessary so that the paper reflects the good experimental work. Here are my comments about it:
Line 52: Regarding the reduction of animal protein and its replacement by vegetable proteins, it would be interesting to point out some of the works that include legumes as a source of protein in snacks such as:
Arribas, C., Cabellos, B., Cuadrado, C., Guillamón, E., & Pedrosa, M. M. (2019). Extrusion effect on proximate composition, starch and dietary fibre of ready-to-eat products based on rice fortified with carob fruit and bean. LWT, 111, 387–393. https://doi.org/10.1016/j.lwt.2019.05.064
Ciudad-Mulero, M., Fernández-Ruiz, V., Matallana-González, M. C., & Morales, P. (2019). Dietary fiber sources and human benefits: The case study of cereal and pseudocereals. Advances in Food and Nutrition Research, 90, 83–134. https://doi.org/10.1016/bs.afnr.2019.02.002
Answer: According to the suggestion, the references have been added (lines 59, 227).
Figure 1: please, include in the graph the bars of the standard deviation of determinations.
Answer: According to the suggestion, we included the bars of the standard deviation in Figure 1.
- Table 2: Perhaps it would be interesting to also include the energy intake of a bar and include the % that covers daily needs (at least in fiber). This information can be included in the text as well.
Answer: According to the suggestion, the required information have been added to the Table 2 (totally revised) and commented in the text (lines 270-272; 282-284; 292-293).
- Perhaps more emphasis could be placed on the correlation between the values ​​evaluated technologically and the characterization made by the experts.
Answer: According to the suggestion, the correlation between instrumental and sensorial analysis has been emphasized (lines 360-365).
